

# Smoothly vanishing density in the contact process by an interplay of disorder and long-distance dispersal

Róbert Juhász[1*], István A. Kovács[2,3,4†]

**1** HUN-REN Wigner Research Centre for Physics, Institute for Solid State Physics and Optics,
H-1525 Budapest, P.O.Box 49, Hungary
**2** Department of Engineering Sciences and Applied Mathematics,
Northwestern University, Evanston, Illinois 60208, USA
**3** Northwestern Institute on Complex Systems,
Northwestern University, Evanston, Illinois 60208, USA
**4** Department of Physics and Astronomy, Northwestern University,
Evanston, Illinois 60208, USA

⋆ juhasz.robert@wigner.hun-ren.hu , † istvan.kovacs@northwestern.edu

## Abstract

Realistic modeling of ecological population dynamics requires spatially explicit descriptions that can take into account spatial heterogeneity as well as long-distance dispersal. Here, we present Monte Carlo simulations and numerical renormalization group results for the paradigmatic model, the contact process, in the combined presence of these factors in both one and two-dimensional systems. Our results confirm our analytic arguments stating that the density vanishes smoothly at the extinction threshold, in a way characteristic of infinite-order transitions. This extremely smooth vanishing of the global density entails an enhanced exposure of the population to extinction events. At the same time, a reverse order parameter, the local persistence displays a discontinuity characteristic of mixed-order transitions, as it approaches a non-universal critical value algebraically with an exponent $\beta_p' < 1$.

# 1 Introduction

Modeling the dynamics of populations is a key challenge in ecology [1]. Beyond traditional mean-field models which consider a single variable (the global density) for each species, a more realistic, although theoretically less tractable class of models is the family of spatially explicit models, which describe the state of the system in terms of local densities in the two-dimensional space. A frequently applied simplification to such spatially explicit models is that local densities are attached to sites of a regular (most frequently square) lattice. A further ingredient of a more realistic modeling beyond mean field is the stochasticity of reproduction and extinction events [2].

The paradigmatic starting point of this kind of modeling is the contact process (CP) [3,4], where the state is given by a set of binary variables at each site, encoding empty (**0**) or colonized (**1**) habitat patches. Colonization of neighboring lattice sites ($\mathbf{0} \to \mathbf{1}$) and spontaneous extinction ($\mathbf{1} \to \mathbf{0}$) occur independently with given rates. Depending on the ratio of these rates, the steady state can be either a completely empty (extinct) state or a fluctuating state with a non-zero global density of colonized sites. In between, a continuous phase transition occurs, which belongs to the directed percolation (DP) universality class [5–7]. As it is common for critical phenomena, the order parameter of the transition, the global density $\rho$, vanishes algebraically with the reduced control parameter $\Delta$ as the transition point is approached from the fluctuating phase:

$$\rho(\Delta) \sim \Delta^{\beta}, \tag{1}$$

$\beta$ denoting the dimension-dependent order-parameter exponent characteristic of the DP universality class.

This simple CP has been extended in at least two directions with the purpose of a more realistic modeling of ecological systems. First, according to observations, colonization events occur also to distant places, and the dispersal is heavy-tailed, i.e., the colonization probability decreases with the distance slower-than-exponentially [8–14]. This issue has been studied in the CP for the case of a dispersal probability decaying algebraically with the distance as $p(l) \sim l^{-\alpha}$ [15–17]. According to these studies, in low dimensions ($d < 4$), the critical behavior remains in the DP class if $\alpha > \alpha_c(d)$, where $\alpha_c(d)$ is a dimension-dependent threshold value, while, for $\alpha_{\mathrm{MF}}(d) < \alpha < \alpha_c(d)$, the critical exponents are altered and depend on $\alpha$. Finally, for $\alpha < \alpha_{\mathrm{MF}}(d) = \frac{3}{2}d$, the system enters the mean-field regime. Thus, even in this case, the algebraic vanishing of the order parameter in Eq. (1) holds to be valid, although with a modified $\beta$. Second, real systems may have a fine-scale heterogeneity in the conditions of surviving or reproduction. This can be taken into account in the CP by quenched random colonization and extinction rates. The well known, perturbative Harris criterion [18] predicts weak disorder to be relevant in low dimensions ($d < 4$), leading to a new type of critical behavior. According to the strong-disorder renormalization group (SDRG) method [19, 20], at least for sufficiently strong disorder,[1] the critical behavior is described by an infinite-disorder fixed point (IDFP) in agreement with results of Monte Carlo simulations [23–25]. At the IDFP, the dynamics is highly uncommon, as time itself is replaced by its logarithm in scaling relations, but otherwise power-laws like Eq. (1) hold to be valid with critical exponents different from those of DP.

Thus, we can see that the most relevant observable of the system, the stationary density $\rho$, follows a power-law scaling with the reduced control parameter given by Eq. (1), even if the model has long-distance dispersal or if it has quenched disorder in it, these circumstances merely affecting the value of $\beta$. Informed by these results, it is natural to expect that the power-law decay of the density holds even in the simultaneous presence of long-distance dispersal

---

[1]At present, it is not known rigorously whether the IDFP is attractive for any weak disorder [21] or there exists a weak disorder regime with a line of fixed points [19, 22].

and quenched disorder. This is, however, not the case. According to SDRG studies of this model, the critical behavior is described by a finite-disorder fixed point (FDFP) at which the dynamical relationship between length and time scales is of power-law type with a logarithmic correction [26–28]. This scenario is expected to be valid in the non-mean-field regime $\alpha > \frac{3}{2}d$, where the Harris criterion predicts weak disorder to be relevant. The predictions of this theory for the dynamical scaling at the critical point have been confirmed by Monte Carlo simulations in one and two dimensions [27, 29]. Furthermore, according to a recent SDRG study of the one-dimensional model by one of us [30], the density vanishes near the critical point according to

$$\rho(\Delta) \sim e^{-C/\sqrt{\Delta}}, \tag{2}$$

with $C$ being a nonuniversal positive constant. This type of infinite-order vanishing of the density, corresponding formally to $\beta = \infty$, is different from the conventional power-law singularity in Eq. (1).

In this paper, we first aim to compare this result of the SDRG treatment in one dimension with Monte Carlo simulations. More importantly, we extend the investigations of the order parameter to two dimensions, which is more relevant to ecological modeling, by a numerical application of the SDRG method and by Monte Carlo simulations.

Besides the density order parameter, we also consider the local persistence, which is the probability that a given site is never activated [31–37]. The persistence can be regarded as a reverse order parameter, which is zero in the active phase and non-zero in the inactive one. In the homogeneous model, it vanishes algebraically as the critical point is approached from the inactive phase,

$$\pi(\Delta) \sim (-\Delta)^{\beta_p}, \tag{3}$$

with a dimension-dependent exponent $\beta_p$. In the simultaneous presence of quenched disorder and long-range interactions, the SDRG method predicts again an anomalous behavior in one dimension: the persistence remains non-zero at the transition point [29] but approaches its critical value $\pi_0$ algebraically [30] as

$$\pi(\Delta) - \pi_0 \sim (-\Delta)^{\beta_p'}, \tag{4}$$

with $\beta_p' = 1/2$, which is characteristic of mixed-order transitions. In this work, we confirm this law by off-critical Monte Carlo simulations and show that similar behavior appears also in two dimensions by the numerical SDRG method and Monte Carlo simulations.

The paper is organized as follows. The model is introduced in Sec. 2.1, followed by a review of the SDRG method in Sec. 2.2 and the details of the Monte Carlo simulation in Sec. 2.3. The numerical results are presented in Sec. 3 and discussed in Sec. 4.

## 2 The model and methods

### 2.1 The contact process

We consider the contact process on a $d$-dimensional lattice (with $d = 1$ or $d = 2$), the sites of which are either active or inactive. The dynamics of the CP is a continuous-time Markov process with two types of independent transitions. First, active sites become inactive with a site-dependent, quenched random rate $\mu_n$. Second, active sites attempt to activate other sites with rates

$$\lambda_{nm} = \Lambda_{nm} r_{nm}^{-\alpha}, \tag{5}$$

where $\Lambda_{nm} = \Lambda_{mn}$ are quenched random rates and $r_{nm}$ denotes the Euclidean distance between the source ($n$) and the target site ($m$). The parameter $\alpha$, which we restrict to the range $\alpha > d$, characterizes the extent of long-distance dispersal.

We study two observables of this model, the density of active sites $\rho$ and the persistence probability $\pi$ at late times. To measure the latter, the system is initiated from a state in which sites are active with a probability $\rho_0 < 1$, and the persistence $\pi(t)$ at time $t$ is defined as the fraction of sites which remained inactive all the way up to time $t$.

## 2.2 The SDRG method

In the SDRG procedure, the maximum $\Omega$ of rates present in the system and the number of effective sites is reduced recursively, step-by-step [19]. In each elimination step, a block of sites containing the largest rate is replaced by simpler blocks. As the model has two sets of rates, there are two kinds of elimination steps. If the largest rate is an activation rate, $\Omega = \lambda_{nm}$, and neighboring deactivation rates are much smaller than $\Omega$, the sites $n$ and $m$ are replaced by a single effective site (cluster) which has an effective deactivation rate

$$\tilde{\mu}_{nm} = \kappa \frac{\mu_n \mu_m}{\Omega}, \tag{6}$$

with $\kappa = 2$ in leading order of perturbation calculation. If the largest rate is a deactivation rate $\Omega = \mu_n$, and the neighboring activation rates are much smaller than this, the site is eliminated, leaving behind effective transitions between all original neighbors of site $n$, with rates obtained perturbatively in the form

$$\tilde{\lambda}_{mk} = \frac{\lambda_{mn} \lambda_{nk}}{\Omega}. \tag{7}$$

When these operations are attempted in $d > 1$ dimensions, one encounters a difficulty of arising double activation rates between clusters. We treated this problem by following the standard 'maximum rule', which means that only the larger one of the rates is kept. A further modification in the rule in Eq. (6) we have done is that $\kappa = 1$ has been used instead of $\kappa = 2$. The reason for this is that this modified rule, together with the maximum rule allows us to use an efficient numerical algorithm of the SDRG procedure developed by one of us [38]. This implementation of the SDRG method works in a parallel manner, relying on graph algorithms to obtain equivalent results to the above mentioned conceptual picture, but in nearly linear time as a function of the number of sites. For the CP with nearest-neighbor dispersal on low-dimensional lattices, where the critical fixed point of the SDRG transformation is an IDFP, both the maximum rule and the irrelevance of the value of $\kappa$ are justified [39]. For the CP with long-distance (algebraic) dispersal, where the critical fixed point is an FDFP [26], Monte Carlo results on the dynamical scaling at the critical point in $d = 1$ and $d = 2$ confirmed the validity of these approximations for $\alpha > \frac{3}{2}d$, and also at $\alpha = \frac{3}{2}d$ for a high enough dilution [27, 29].

The density $\rho(t)$ of active sites and the persistence probability $\pi(t)$ can be calculated within the SDRG approach in the following way. The density $\rho(t)$ at time $t$, if initially each site was active, is given by the ratio of those sites which are part of an active (not yet decimated) cluster at rate scale $\Omega = t^{-1}$. The density in the quasistationary state of a finite system in the active phase is thus given by the ratio of sites contained in the last surviving cluster formed in the SDRG procedure. The persistence of a given site $(n)$ can be obtained as follows [31]. The deactivation rate at this site is set to zero, $\mu_n = 0$, then this site remains persistent as long as it is not merged with another cluster during the SDRG transformation. Performing this procedure for all sites and counting the fraction of persistent sites at $\Omega = t^{-1}$ provides $\pi(t)$.

In the numerical calculations, the rates $\Lambda_{nm}$ and $\mu_n$ were drawn from uniform distributions in the interval $(0, 1)$ and $(0, \mu)$, respectively. As a control parameter, we used $\Theta = \ln \mu$. For the $d = 2$ model, we set the dispersal exponent to $\alpha = 3$, which tests the range of validity ($\alpha \geq \frac{3}{2}d$) of the SDRG method. In this case, the critical point was found to be in an earlier work at $\Theta_c = 2.42(5)$ [27] by analysing the distribution of sample-dependent critical points [26]. We considered different system sizes up to $L = 64$ and used typically 40 000 random samples (and at least 4000 for the largest size of the density calculation).

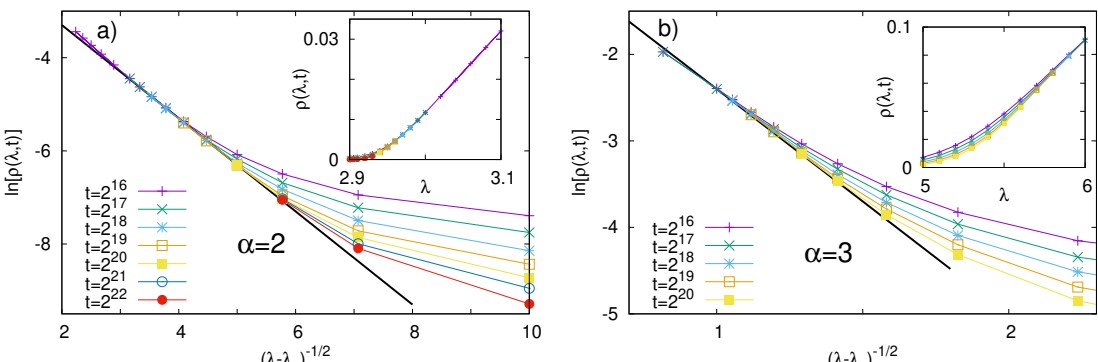

Figure 1: Dependence of the global density on the control parameter $\lambda$ at different times in the one-dimensional model with $\alpha = 2$ (a) and $\alpha = 3$ (b). Data have been obtained by Monte Carlo simulations. The critical control parameters are $\lambda_c = 2.90(1)$ for $\alpha = 2$ and $\lambda_c = 5.00(5)$ for $\alpha = 3$ [27]. The straight lines are linear fits to the stationary data having slopes $-C = -1.0$ (a) and $-C = -2.6$ (b).

## 2.3 Monte Carlo simulation

In the simulations, a different type of disorder, which is more frequently used in Monte Carlo studies [24,25,27], was considered. This is a random dilution of the lattice, by which a fraction $c$ of sites is randomly deleted. For practical reasons, it is expedient to use high values of $c$, as low dilutions induce an undesirable crossover from the critical behavior of the clean system, shrinking the true asymptotic behavior to the close vicinity of the critical point. Note that, due to long-distance dispersal, $c$ does not even need to be below the percolation threshold of the underlying regular lattice.

We applied a sequential update, in which an active site is picked randomly and either made inactive with a probability $1/(1 + \lambda)$ or an activation event is attempted from this site with a probability $\lambda/(1 + \lambda)$. In the latter case, a target site is selected randomly with a probability proportional to $l^{-\alpha}$, where $l$ is the Euclidean distance measured from the source site. As control parameter of the transition we used $\lambda$. Technical details of the simulation are the same as in Ref. [27].

Measurements of the global density $\rho(t)$ were performed in a single large sample for each value of the control parameter. The size of the sample was $L = 10^9$ in one dimension and the linear size was $L = 30\,000$ in two dimensions. Measurements of the persistence in the inactive phase were performed by starting the process from a random state with a global density $\rho_0 = 1/2$, simulating until the system reached the fully inactive (absorbing) state. The fraction of persistent sites was then calculated and averaged typically over results obtained in 100 random samples. The maximal linear system size was $L = 10^8$ for $d = 1$ and $L = 40\,000$ for $d = 2$.

# 3 Numerical results

## 3.1 The one-dimensional model

The one-dimensional model has been studied by a simplified, analytically tractable SDRG scheme which contained a series of approximations [26,30]. First, at any stage of the renormalization, the only interactions that were taken into account were those between adjacent clusters. Furthermore, these couplings were approximated by the long-range interaction between the closest constituent sites. Up to this point, the scheme is essentially equivalent to

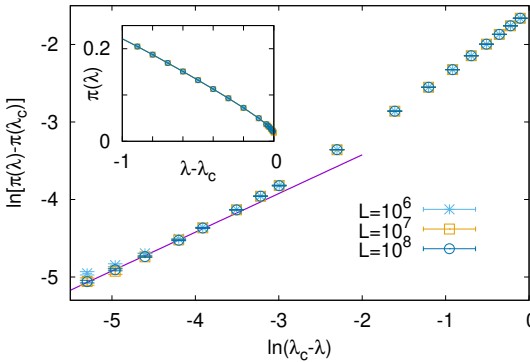

Figure 2: Dependence of the persistence probability on the control parameter for different system sizes in the one-dimensional model. Data have been obtained by Monte Carlo simulations with $\alpha = 2$ and $c = 0.5$. The straight line has a slope 0.499.

the application of the maximum rule. Second, the extension of clusters was neglected compared to the spacings between them. This step is justified in the inactive phase and in the active phase close to the critical point. This simplified method predicted a smooth vanishing of the global density in the active phase as given by Eq. (2) [30]. To check the validity of this asymptotic form, we performed Monte Carlo simulations for $\alpha = 2$ and $\alpha = 3$, with a dilution parameter $c = 0.5$. Data obtained for different values of the control parameter $\lambda$ and at different times are plotted in Fig. 1. Plotting the logarithmic densities $\ln \rho(\lambda, t)$ against $\Delta^{-1/2}$, with $\Delta = \lambda - \lambda_c$, leads to an agreement with Eq. (2), as the late-time densities (which approached their steady-state values) fit well to a straight line. Here, estimates of the critical control parameter $\lambda_c$ were taken from Ref. [27]. We find the constant $C$ appearing in Eq. (2) to vary with $\alpha$.

Next, we measured the persistence probability, which has been found by the SDRG method to exhibit a mixed-order type of transition given by Eq. (4) [30]. Numerical results obtained by Monte Carlo simulations in the inactive phase for $\alpha = 2$, $c = 0.5$, are plotted in Fig. 2. Using the critical persistence $\pi(\lambda_c) = 0.015$ estimated in Ref. [29], we find that data are compatible with the form given by Eq. (4) and the estimate 0.499 of the exponent $\beta'_p$ obtained by a linear fit to the data on a double logarithmic scale close to the critical point provides a good agreement with the theoretical value $\beta'_p = 1/2$.

## 3.2 The two-dimensional model

Unlike in one dimension, the SDRG method is not analytically tractable in higher dimensions, thus we have no analytical prediction about the scaling of order parameters at our disposal in two dimensions. Nevertheless, by a relatively simple argument, a relationship between the models in different dimensions was established in Ref. [27]. This relies on a recursive rearrangement of the sites of the $d$-dimensional model to a one-dimensional chain, after which distances $l$ between sites of the $d$-dimensional model are typically scaled to $l^d$. As a consequence, the critical behavior of the $d$-dimensional model having a dispersal parameter $\alpha$ is expected to be the same as that of the one-dimensional model with a reduced dispersal parameter $\alpha/d$. Thereby, formulae known for the one-dimensional model can be extended to $d > 1$ by replacing $\alpha$ by $\alpha/d$, as well as all distances by their $d$th power. The precision of this statement is presumably restricted to the equality of critical exponents and may not hold for any additional multiplicative logarithmic corrections [27]. Numerical SDRG and Monte Carlo analyses of the dynamical scaling of the order parameter at criticality in two and three dimensions have supported this general connection [27, 28].

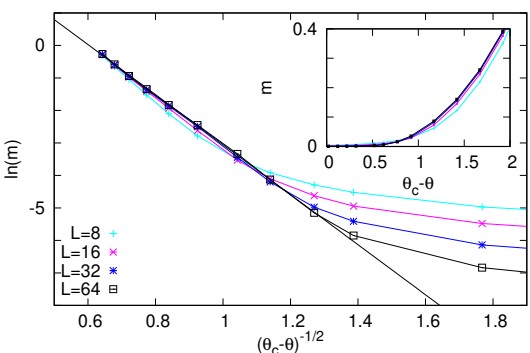

Figure 3: The order parameter $m$ obtained by the numerical SDRG method in the two-dimensional model with $\alpha = 3$, plotted against the reduced control parameter $\Delta = \Theta_c - \Theta$ (inset). The critical control parameter is $\theta_c = 2.42(5)$. In the main panel, the same data are linearized according to Eq. (2). The straight line has a slope $-C = -7.7$.

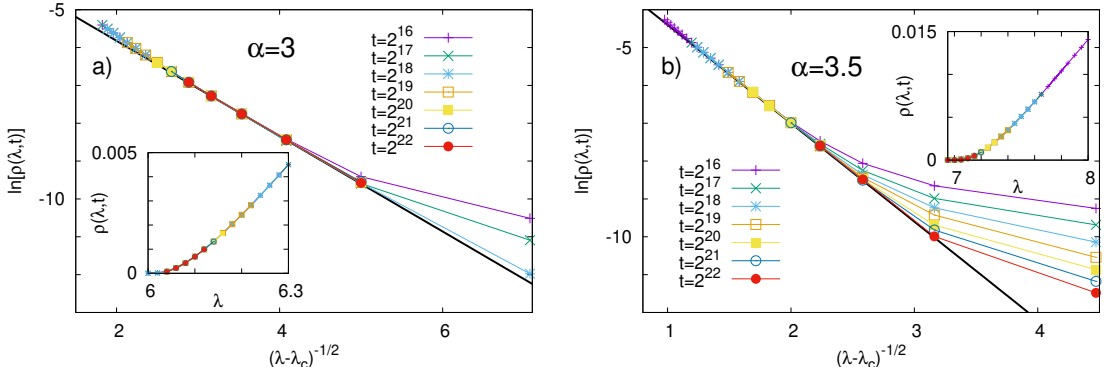

Figure 4: Dependence of the global density on the control parameter $\lambda$ at different times in the two-dimensional model with $\alpha = 3$ (a) and $\alpha = 3.5$ (b). Data have been obtained by Monte Carlo simulations. The critical control parameters are $\lambda_c = 6.00(2)$ for $\alpha = 3$ and $\lambda_c = 5.95(5)$ for $\alpha = 3.5$ [27]. The straight lines are linear fits to the stationary data having slopes $-C = -1.26$ (a) and $-C = -2.60$ (b).

Applying this conjecture to the off-critical scaling of the density in Eq. (2), as it contains neither distances nor the parameter $\alpha$, leads to the expectation that it holds in the same form also for $d > 1$. Therefore, our numerical analysis in two dimensions was guided by this form. First, we performed a numerical SDRG analysis and measured the fraction of active (non-decimated) sites $m$ contained in the last cluster, a quantity which is expected to display the same scaling behavior as the global density. Numerical results for $\alpha = 3$ are shown in Fig. 3. As shown in the inset of Fig. 3, the order parameter vanishes smoothly as the critical point is approached, and, indeed, the main panel of Fig. 3 shows that the data fit well to the form given by Eq. (2).

Results on the global density obtained by Monte Carlo simulations of the two-dimensional model with a dilution parameter $c = 0.8$ and dispersal exponents $\alpha = 3$ and $3.5$ are shown in Fig. 4. Here, we used again estimates of the location of critical point ($\lambda_c = 6.00(2)$ for $\alpha = 3$ and $\lambda_c = 6.95(5)$ for $\alpha = 3.5$) from an earlier work [27]. As shown in the figure, the stationary data fit well to the form in Eq. (2) for both values of $\alpha$, and the constant $C$ is different for different $\alpha$.

Next, we turned to the dependence of persistence probability on the control parameter in the inactive phase. The conjectured connection between the models in different dimensions

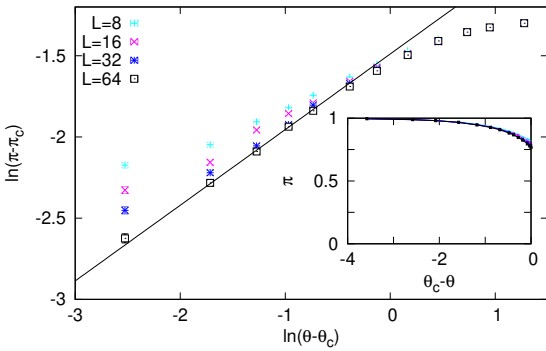

Figure 5: The persistence $\pi$ obtained by the numerical SDRG method in the two-dimensional model with $\alpha = 3$, plotted against the reduced control parameter $\Delta = \Theta_c - \Theta$ (inset). The main panel shows the same data linearized according to Eq. (4). The straight line has a slope $\beta'_p = 0.47$.

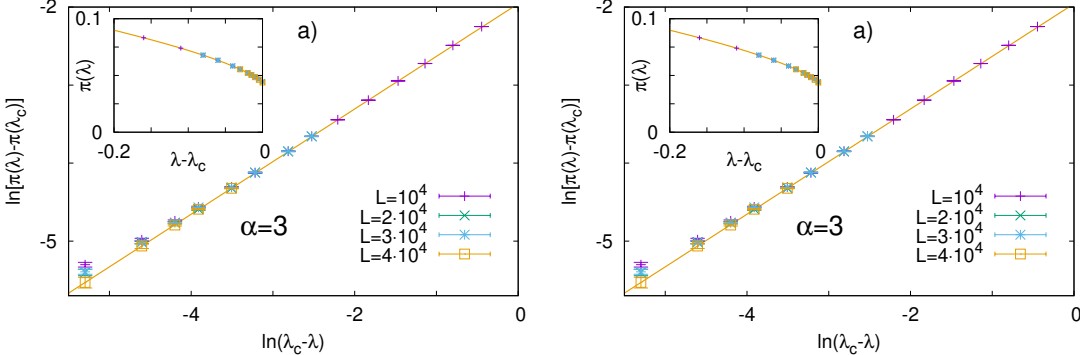

Figure 6: Dependence of the persistence probability on the control parameter for different system sizes. Data have been obtained by Monte Carlo simulations of the two-dimensional model with $\alpha = 3$ (a) and $\alpha = 3.5$ (b). The straight lines have slopes $\beta'_p = 0.69$ (a) and $\beta'_p = 0.68$ (b).

described above would naïvely predict that a relationship given by Eq. (4) holds to be valid also for $d > 1$. The question is, however, which one of its parameters, if any, remains unchanged when switching to higher dimensions. The leading (constant) term $\pi_0$ is non-universal, i.e., it depends on the type of disorder and $\alpha$, therefore it is not expected to remain unchanged. The invariance of the exponent $\beta'_p$ is also questionable, as it appears in the subleading term. Therefore, in the numerical analysis, we assumed a mixed-order type of form for the vanishing of the persistence where $\pi_0$ and $\beta'_p$ are treated as fitting parameters.

Numerical results in the inactive phase obtained by the SDRG method for $\alpha = 3$ are shown in Fig. 5. As illustrated, the data are compatible with a mixed-order vanishing and the estimates of the parameters obtained by a fitting to the data for the largest system size ($L = 64$) are $\pi_0 = 0.72(1)$ and $\beta'_p = 0.47(4)$. Nevertheless, care has to be taken with these estimates since, as we can see in the figure, the persistences close to the critical point are still considerably affected by the finite size of the system. Thus, the asymptotic ($L \to \infty$) value of $\pi_0$ may be significantly lower, while $\beta'_p$ may be significantly higher than the estimates obtained for $L = 64$.

Results of Monte Carlo simulations obtained for $\alpha = 3$ and $\alpha = 3.5$ (and a dilution parameter $c = 0.8$) are shown in Fig. 6. These confirm what has been found by the SDRG method: the vanishing is of mixed-order type. The estimated critical persistences are $\pi_0 = 0.043(2)$ for $\alpha = 3$ and $\pi_0 = 0.070(2)$ for $\alpha = 3.5$. Since the value of $\beta'_p$ obtained by fitting is very

sensitive to the errors of $\lambda_c$ and $\pi_0$, we have rather uncertain estimates on it: $\beta'_p = 0.7(2)$ both for $\alpha = 3$ and $\alpha = 3.5$. Note that, just like the two-dimensional SDRG results, these do not exclude a possible agreement with the value $1/2$ obtained by the SDRG method for $d = 1$.

# 4 Discussion

In this work, we considered a stochastic lattice model of population dynamics, the contact process in the simultaneous presence of quenched disorder and long-distance dispersal. We focused on the off-critical scaling of order parameters, for which an earlier SDRG work predicted anomalous scaling behavior in one dimension, for large enough dispersal exponents ($\alpha \geq \frac{3}{2}d$). Specifically, the global density vanishes smoothly, whereas the persistence probability vanishes discontinuously when the critical point is approached from the active phase and the inactive phase, respectively [30]. We provided a numerical confirmation of these forms by Monte Carlo simulations in one dimension and, making a step toward a more realistic modelling, we addressed this issue in the two-dimensional model. In two dimensions, our numerical SDRG and Monte Carlo results agree in that the global density follows the same type of smooth vanishing given in Eq. (2) as found in one dimension. Furthermore, the discontinuous, mixed-order type of vanishing of the persistence found in one dimension holds to be valid also in two dimensions, as it is demonstrated by our numerical SDRG and Monte Carlo results. Concerning the exponent $\beta'_p$ in the subleading term of persistence, our numerical results clearly signal a singularity with $\beta'_p < 1$, but the accessible numerical data are not conclusive on whether this exponent is different from that of the one-dimensional model ($1/2$) or not.

We note that the compatibility of SDRG and Monte Carlo results suggests that the SDRG method is a valid approach also for the random transverse-field Ising model with long-range interactions [27, 28]. For that model, the SDRG scheme is formally similar to that of the CP (with $\kappa = 1$), and, as a consequence, the magnetization (corresponding to the global density of CP) is expected to vanish according to Eq. (2) as the quantum critical point is approached from the ferromagnetic phase at zero temperature.

The type of extremely smooth vanishing of the density at the extinction threshold characterized formally by $\beta = \infty$ is in stark contrast to the linear vanishing of the density in the mean-field theory ($\beta_{MF} = 1$) and, especially, with the singular vanishing of the density in the $d = 2$ DP universality class, where $\beta = 0.583(4)$ [5]. Remarkably, in the latter case, the density tends to zero at the critical point with an infinite slope, while in the model studied in this work, with a zero slope. We note, that the two-dimensional disordered CP with a short-range dispersal, for which $\beta = 1.23(4)$ [38], also displays a zero-slope vanishing similar to the present model. Nevertheless, unlike in our model, the order-parameter exponent is finite there, and is just barely greater than 1. The extremely smooth vanishing of the global density may have a profound impact on the expected lifetime and extinction of finite populations. In case of a finite exponent $\beta$, in particular for $\beta < 1$, the population may have a low risk of extinction even fairly close to the extinction threshold due to the relatively high global density there. This is, however, not the case for $\beta = \infty$, where the mean density is rather low even well above the threshold, and, as a consequence, the population is more easily exterminated by demographic or environmental fluctuations.

There are several directions in which the investigations performed in this work could be extended. From a theoretical perspective, one may pose the question whether the anomalous behavior of order parameters found for $d = 1$ and $d = 2$ holds to be valid in higher dimensions, as well. Based on the conjectured connection between different dimensional variants of the model, we expect qualitatively similar results in three dimensions (with a possibly dif-

ferent value of $\beta'_p$) in the non-mean-field domain $\alpha > \frac{9}{2}$, where the Harris criterion predicts weak disorder to be relevant. However, in four dimensions and above, where weak disorder is irrelevant for any value of the dispersal exponent $\alpha$, just as for $\alpha < \frac{3}{2}d$ in lower dimensions, we expect the model to exhibit mean-field critical behavior characterised with $\beta_{\mathrm{MF}} = 1$. With the scope of ecological modeling, it would be interesting to analyse the model under an environmental gradient, which is typically realized for plant species living on a hillside, where the environmental factors become more and more unfavorable with increasing or decreasing altitude [40]. In modeling, this circumstance can be implemented by applying a linear spatial trend in the rates along a given direction. Intuitively, such trends are expected to transform the control-parameter dependence of global density examined in this paper to a spatial dependence of the local density along the direction of the gradient. Moreover, the study of environmental gradients would allow us to confront the predictions of the model with observational data accessible by satellite images on the spatial variation of the local density of a given species near the range margin.

## Acknowledgments

**Funding information** This work was supported by the National Research, Development and Innovation Office NKFIH under Grant No. K146736. The work of IAK was supported by the National Science Foundation under Grant No. PHY-2310706 of the QIS program in the Division of Physics.

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
