# Peer review of "Smoothly vanishing density in the contact process by an interplay of disorder and long-distance dispersal"

_SciPost Physics Core, doi:SciPost Phys. Core 7, 044 (2024)_

## Round 2 · Referee Report · Anonymous (Referee 1) · 2024-6-16

Report

The authors have replied in a satisfactory way to the minor comments in my previous report, and made the related changes in the manuscript. I therefore support the publication of this work.

Recommendation

Publish (meets expectations and criteria for this Journal)

---

## Round 2 · Author Response

Dear Editor,

We revised the manuscript following the Reviewer's comments. Please, find our responses below.

Sincerely yours,

Róbert Juhász
István Kovács

---

## Round 2 · List of Changes

Reply to the Reviewer

Reviewer:
I think that the presentation of certain results could be slightly improved, to make the manuscript more self-consistent. My suggestions in this direction are reported below.

Response: We thank the Reviewer for the careful assessment of our manuscript and for the constructive comments.

Reviewer:
(i) The Authors stress that the infinite-order behavior in Eq. (2) is a consequence of the combined presence of quenched randomness in the local rates and long-range dispersal, while the two ingredients taken separately lead to an algebraic vanishing of the density at the extinction transition. Do the Authors have some insight on the mechanism behind this?

Response: The difference is in the functional form of the vanishing of the order parameter, so it is at most a quantitative difference. The physics is essentially (qualitatively) the same as in the short-range disordered model. The long-range interaction induces a change in the functional form, which is an outcome of the renormalization method. Unfortunately, we are not aware of any heuristic picture which could capture such a quantitative change.

Reviewer:
(ii) The simplified, analytically tractable SDRG analysis of Ref. [31] is mentioned in several points of the manuscript: I would suggest to briefly describe what are the assumptions behind that simplified treatment, in order to make the manuscript self-contained. I would do the same regarding the numerical algorithm to perform SDRG mentioned on p.3 and introduced in Ref. [39].

Response: We have added more detail to both sections (II.B and III.A) on these methods to emphasize the assumption behind the simplified treatment as well as the essence of the efficient SDRG algorithm.

Reviewer:
(iii) What justifies the choice of c, the dilution parameter, used in the Monte Carlo numerics?

Response: The choice of c is arbitrary but there are practical aspects, which we commented on in Sec. II.C. as follows:
"For practical reasons, it is expedient to use high values of $c$, as low dilutions induce an undesirable crossover from the critical behavior of the clean system, shrinking the true asymptotic behavior to the close vicinity of the critical point."

Reviewer:
(iv) Could the Authors comment on what are the expectations in the case of truly long-range dispersal, i.e. when the exponent alpha is smaller than the dimensionality d?

Response: We added a comment on this point to the Discussion as follows:
"...just as for $\alpha<\frac{3}{2}d$ in lower dimensions, we expect the model to exhibit mean-field critical behavior characterised with $\beta_{\rm MF}=1$."

Reviewer:
(v) In the discussion, the Authors mention the model with an environmental gradient as a perspective, and refer to comparison with satellite image data; I would add some reference or clarify this context: which type of data should one think of, which one hopes to describe in terms of a contact process? How should the environmental gradient be implemented?

Response: We explained this point in more detail at the end of the Discussion as follows:
"...an environmental gradient, which is typically realized for plant species living on a hillside, where the environmental factors become more and more unfavorable with increasing or decreasing altitude \cite{oborny}. In modeling, this circumstance can be implemented by applying a linear spatial trend in the rates along a given direction. ... Moreover, the study of environmental gradients would allow us to confront the predictions of the model with observational data accessible by satellite images on the spatial variation of the local density of a given species near the range margin."

---

## Editorial Decision

published